# MiR-155-5p Attenuates Vascular Smooth Muscle Cell Oxidative Stress and Migration via Inhibiting BACH1 Expression

**DOI:** 10.3390/biomedicines11061679

**Published:** 2023-06-09

**Authors:** Ying Tong, Mei-Hui Zhou, Sheng-Peng Li, Hui-Min Zhao, Ya-Ru Zhang, Dan Chen, Ya-Xian Wu, Qing-Feng Pang

**Affiliations:** 1Department of Physiopathology, Wuxi School of Medicine, Jiangnan University, 1800 Lihu Avenue, Binhu District, Wuxi 214122, China; 2Department of Pathophysiology, Nanjing Medical University, Nanjing 211166, China

**Keywords:** vascular smooth muscle cells, migration, miR-155-5p, BACH1, oxidative stress

## Abstract

The malfunction of vascular smooth muscle cells (VSMCs) is an initiating factor in the pathogenesis of pathological vascular remodeling, including hypertension-related vascular lesions. MicroRNAs (miRNAs) have been implicated in the pathogenesis of VSMC proliferation and migration in numerous cases of cardiovascular remodeling. The evidence for the regulatory role of miR-155-5p in the development of the cardiovascular system has been emerging. However, it was previously unclear whether miR-155-5p participated in the migration of VSMCs under hypertensive conditions. Thus, we aimed to define the exact role and action of miR-155-5p in VSMC migration by hypertension. Here, we detected that the level of miR-155-5p was lower in primary VSMCs from spontaneously hypertensive rats (SHRs). Its overexpression attenuated, while its depletion accelerated, the migration and oxidative damage of VSMCs from SHRs. Our dual-luciferase reporter assay showed that miRNA-155-5p directly targeted the 3′-untranslated region (3′-UTR) of BTB and CNC homology 1 (BACH1). The miR-155-5p mimic inhibited BACH1 upregulation in SHR VSMCs. By contrast, the deletion of miR-155-5p further elevated the upregulation of BACH1 in SHR-derived VSMCs. Importantly, the overexpression of miR-155-5p and knockdown of BACH1 had synergistic effects on the inhibition of VSMCs in hypertension. Collectively, miR-155-5p attenuates VSMC migration and ameliorates vascular remodeling in SHRs, via suppressing BACH1 expression.

## 1. Introduction

Pathophysiological vascular remodeling is characterized by vascular cell proliferation, apoptosis, and migration, which is critically involved in increased peripheral resistance, thereby leading to the development of hypertension [1,2]. Accordingly, vascular remodeling can be reversed to prevent or treat diseases arising from hypertension [3,4]. The excessive migration of vascular smooth muscle cells (VSMCs) is an important pathogenic event in several vascular diseases, including hypertension-induced vascular remodeling [5,6]. The targeting of VSMC migration might function as a promising therapeutic strategy for hypertension and its associated vascular remodeling [7].

It is known that microRNAs (miRNAs) are a class of small non-coding RNA molecules that bind to the 3′-untranslated regions (3′-UTRs) of their target mRNAs to induce translation repression or mRNA degradation [8]. Mature miRNAs can be processed from the 5′ or 3′ arms of miRNA precursors (pre-miRNAs) to form miR-5p and miR-3p [9]. The knockdown of miR-155-5p promotes the proliferation and migration of human brain microvessel endothelial cells [10]. The role of miR-155-5p in the production of vascular inflammatory cytokines has also been established [11]. The vascular mineralocorticoid receptor regulates miR-155-5p to affect vasoconstriction and blood pressure in mice [12]. In addition, miR-155-5p deficiency attenuates vascular calcification by reducing VSMC apoptosis [13]. Our group has shown that miR-155-5p in adventitial extracellular vesicles originating from fibroblasts inhibits VSMC proliferation in hypertension [14], and that miR-155-5p inhibits cell migration and oxidative stress in the VSMCs of spontaneously hypertensive rats [15]. However, the potential role and mechanism of miR-155-5p in VSMC migration have yet to be fully elucidated. Thereafter, the focus of the current study was to elaborate the roles of miR-155-5p in VSMC migration in hypertension, and the underlying mechanism.

## 2. Materials and Methods

Male Wistar–Kyoto rats (WKYs; *n* = 10) and spontaneously hypertensive rats (SHRs; *n* = 10) (aged 8 weeks) were provided by Vital River Laboratory Animal Technology Co. Ltd. (Beijing, China), and were prepared for the isolation of the primary rat aortic vascular smooth muscle cells (VSMCs). For the Western blot and RT-PCR tests, the samples were randomly obtained from four WKY-derived VSMCs, and four SHR-derived VSMCs. For other experiments, the samples were randomly obtained from six WKY-derived VSMCs, and six SHR-derived VSMCs. All animal experiments observed the Guide for the Care and Use of Laboratory Animals (NIH, 8th edition, 2011), and were approved by the Experimental Animal Care and Use Committee of Nanjing Medical University (IACUC No:2107007). In brief, the thoracic aorta was separated (with the perivascular adipose tissues stripped away), cut open to dissect the intima, challenged with 0.4% type 1A collagenase in PBS for digestion for 30 min at 37 °C, and centrifuged. Then, the isolated cells were resuspended in DMEM containing 10% fetal bovine serum, 100 IU/mL penicillin, and 10 mg/mL streptomycin, and maintained with 5% CO_2_ at 37 °C. The VSMCs between the second and the sixth passages were used in this study.

### 2.1. BACH1 Knockdown in VSMCs

BACH1-siRNA-lentivirus (1 × 10^9^ TU/mL) was designed by Generay Biotech Co., Ltd. (Shanghai, China). The sequence of the BACH1-siRNA-lentivirus nucleotide was 5′-GGAACCGACAAGATCCGAACT-3′. We used BACH1-siRNA-lentivirus (MOI = 80) to infect the VSMCs, and scrambled siRNA as a negative control. Measurements were performed after 48 h. The infection efficiency was evaluated using a fluorescence microscope (Axio Vert. Al, Zeiss, Jena, Germany).

### 2.2. Transfection of miR-155-5p Mimic and Inhibitor

Scrambled sequences of the miR-155-5p mimic and inhibitor were used as negative controls to exclude the impact of the reagent or transfection protocol on the target protein. The VSMCs (about 5 × 10^5^ cells/well) were subjected to an 18 h culture in 6-well plates, transfected with miR-155-5p mimics (50 nmol/L) or inhibitors (100 nmol/L), or their related negative controls containing the RNAifectin™ transfection reagent (6 μL). Six hours later, the culture medium was replenished, the transfection reagent discarded, and the transfection efficiency after 24 h evaluated. The RNAifectin™ transfection reagent, miR-155-5p mimics, inhibitors, and their negative controls were designed by Applied Biological Materials Inc. (Richmond, BC, Canada).

### 2.3. ROS Measurement

Cells were stained with 5 μM DCFH-DA (Beyotime, Shanghai, China) for 30 min at 37 °C. Having been washed thrice with PBS, the cells were imaged using the Olympus DP70 digital camera and the Olympus BX51 microscope (Olympus, Tokyo, Japan).

### 2.4. NAD(P)H Oxidase Activity Assay

The activity of NAD(P)H oxidase was measured using commercial kits (Abcam; Cambridge, MA, USA). At 450 nm, the optical density was determined using Microplate Reader (STNERGY/H4, BioTek, Winooski, VT, USA).

### 2.5. VSMC Migration Assay

The migration ability of VSMCs was measured in the Boyden chamber assay and wound-healing assay. In the Boyden chamber assay, the VSMCs were planted into a serum-free medium in the upper 24-well transwell chamber with 8 μm pores (Merck kGaA, Darmstadt, Germany). Then, 10% FBS was injected into the lower chamber, with the cellular migration measured 24 h later. The VSMCs that had not migrated from the upper chamber were scratched using cotton swabs, and those that had migrated into the lower chamber were subjected to 1% crystal violet staining. Five fields were randomly selected from each chamber, and the cells were counted. In the wound-healing assay, the VSMCs (2 × 10^5^ cells/mL) were seeded into 6-well plates, and cultured for 24 h until reaching an 80–90% confluence. Then, the cells were scratched with the tip of a standard 1 mL pipette, until a wound was observed in the central area. Once the cellular debris was washed using PBS, the medium was freshly replenished. Photos of the healing process were captured at 0 and 24 h, using an inverted microscope (Axio Vert. A1, Zeiss, Oberkochen, Germany). The distances among migrated cells were photographed and figured.

### 2.6. Dual-Luciferase Reporter Assay

The VSMCs at a confluence of 85% to 90% were co-transfected with 1 µg/mL pcDNA-BACH1 reporter plasmids prepared by Generay Biotech Co., Ltd. (Shanghai, China). Then, the cells were incubated with normal control and miR-155-5p mimics (50 nmol/L) containing the Lipofectamine™3000 transfection reagent. A dual-luciferase reporter assay system was introduced to evaluate the luciferase activity.

### 2.7. Western Blot Analysis

After the aortic tissue samples were homogenized in an ice-cold lysis buffer, the supernatant was extracted with a BCA protein assay kit (BCA; Pierce, Santa Cruz, CA, USA). This was followed by isolating the total protein using sodium dodecyl sulfate–polyacrylamide gel electrophoresis (SDS-PAGE), and blotting with a polyvinylidene fluoride (PVDF) membrane. Protein bands were quantified using the Enhanced Chemiluminescence Detection Kit (Thermo Fisher Scientific, Rockford, IL, USA). Abcam (Cambridge, MA, USA) provided antibodies against NOX2, NOX4, and BACH1; Cell Signaling Technology (Beverly, MA, USA) provided antibodies against β-actin. Santa Cruz Biotechnology Inc. (Santa Cruz, CA, USA) provided secondary antibodies.

### 2.8. Measurement of the miR-155-5p Level

The total RNA was exacted using the miRcute miRNA isolation kit, quantified using the NanoDrop 2000 Spectrophotometer (Thermo Fisher Scientific, Wilmington, DE, USA), and reverse-transcribed to cDNA using the miRcute Plus miRNA First-Strand cDNA kit (TIANGEN Biotech) for miRNA. The RT-PCR for miR-155-5p was measured using a commercial kit containing primers for miR-155-5p and U6 (TIANGEN Biotech, Beijing, China). The endogenous control used for the miRNA normalization was U6 small nuclear. The primers were found to be compliant with the MIQE guidelines, based on their Ct values determined with a standard curve.

### 2.9. Real-Time PCR

The total RNA extraction was performed using Trizol reagent (Life Technologies, Gaithersburg, MD, USA). At 260 and 280 nm, the optical density was calculated, to assess the RNA concentrations and purity. For the reverse transcriptase reactions, we introduced the PrimeScript^®^ RT reagent Kits (Takara, Otsu, Shiga, Japan) and ABI PRISM 7500 sequence detection PCR system (Applied Biosystems, Foster City, CA, USA). The level of protein expression was normalized to that of GAPDH. The sequences of primers are shown in the Online-only Data Supplement (Appendix A).

### 2.10. Statistical Analysis

Data are presented as mean ± SE. Multiple comparisons were accomplished through one-way or two-way ANOVA, with post hoc Bonferroni testing as appropriate. *p* < 0.05 denotes statistical significance.

## 3. Results

### 3.1. Effects of miR-155-5p Overexpression on VSMC Oxidative Stress in WKYs and SHRs

Despite miR-155-5p being a major player in the proliferation and apoptosis of VSMCs during the development of abdominal aortic aneurysms [16], the role of miR-155-5p in VSMC migration and hypertensive vascular remodeling is largely uncharacterized. Wound-healing and transwell assays were thus used to determine the potential role of miR-155-5p in primary VSMCs. Our previous research indicated that miR-155-5p inhibits cell migration in the VSMCs of spontaneously hypertensive rats [15]. However, the potential role and mechanism of miR-155-5p in VSMC migration have yet to be fully clarified.

Oxidative stress is a vital initiating factor in the phenotype transformation of VSMC, and subsequent VSMC proliferation, apoptosis, and migration, in various vascular diseases, including hypertensive vascular remodeling [17,18]. We evaluated whether the genetic manipulation of miR-155-5p had an impact on the oxidative burst in VSMCs from WKYs and SHRs. DCFH-DA staining results showed that the ROS fluorescence intensity was higher in the SHR VSMCs, which was mitigated by the ectopic expression of miR-155-5p, which itself did not affect the VSMC oxidative state in WKYs (Figure 1A,B). The antioxidant effects of miR-155-5p were also confirmed through the measurement of NAD(P)H oxidase activity (Figure 1C). As two isoforms of NAD(P)H oxidase, NOX2 and NOX4 are primary resources for ROSs in cardiovascular cells, and targeting NOX2 and/or NOX4 could be an important strategy for treating cardiovascular diseases [19,20]. The immunoblotting results revealed that miR-155-5p mimic transfection significantly suppressed the protein expression of NOX2 and NOX2 in SHR VSMCs, when compared with scramble transfection, but not in WKY VSMCs (Figure 1D). Conversely, the gene ablation of miR-155-5p exhibited the opposite phenotypes on ROS production in WKY and SHR VSMCs (Figure 2).

### 3.2. MiR-155-5p Targeted BACH1 in VSMCs

BACH1 is a well-known transcriptional repressor of the cytoprotective enzyme heme oxygenase-1 (HO-1), and a deficiency in BACH1 reduces the VSMC proliferation and neointimal formation [21]. In our study, compared to normal VSMCs, the mRNA level of miR-155-5p was lower in SHR VSMCs (Figure 3A). We were intrigued to know whether BACH1 was a direct target of miR-155-5p in VSMCs. The targeting relationships between miR-155-5p and BACH1 were predicted using TargetScan, and confirmed by the dual-luciferase reporter assay. A bioinformatics analysis performed using TargetScan demonstrated that BACH1 contained an miR-155-5p binding site at the 3′-untranslated region (3′-UTR). The dual-luciferase reporter assay revealed that miR-155-5p upregulation downregulated the transcriptional activity of BACH1 in VSMCs (Figure 3B). The mRNA and protein levels of BACH1 tended to be higher in SHR VSMCs, but were diminished by the silencing of BACH1 (Figure 3C). The overexpression of miR-155-5p inhibited the transcriptional and translational levels of BACH1 in VSMCs from both WKYs and SHRs (Figure 4A). The hypertension-induced upregulation of BACH1 in VSMCs was also erased by miR-155-5p transfection (Figure 4A). In sharp contrast, the loss of miR-155-5p displayed a contrary tendency (Figure 4B). These results indicate that miR-155-5p targeted BACH1 to regulate VSMC functions in hypertension.

### 3.3. Ablation of BACH1 on VSMC Migration and Oxidative Damage in WKYs and SHRs

Next, we examined whether the knockdown of BACH1 affected the proliferation and ROS production of VSMCs in hypertension. The migration capabilities of SHR VSMCs were significantly lower in the BACH1-siRNA group (Figure 5). Accordingly, treatment with BACH1-siRNA evidently suppressed the overproduction of ROSs and NAD(P)H activation in SHR VSMCs (Figure 6A–C), along with the downregulation of NOX2 and NOX4 at the mRNA and protein levels (Figure 6D).

### 3.4. The Ablation of BACH1 Facilitated the Inhibitory Effects of miR-155-5p Mimics on VSMC Migration and ROS Formation in SHRs

Given that miR-155-5p could inhibit VSMC migration and superoxide generation under hypertensive conditions by downregulating BACH1 expression, it is unclear whether BACH1 deficiency and miR-155-5p overexpression played a synergistic role in the regulation of VSMC behaviors in SHRs. Working in tandem, the silencing of BACH1 and the upregulation of miR-155-5p repressed the migration of VSMCs from SHRs, as indicated by the wound-healing and transwell assays (Figure 7). In keeping with this, the downregulation of BACH1 and upregulation of miR-155-5p were able to abate ROS generation and NOX2/4 activation in SHR VSMCs, and they also showed coordinative effects (Figure 8).

## 4. Discussion

In this study, we provide ample evidence that the level of miR-155-5p was diminished in SHR VSMCs, and that miR-155-5p mimics prevented the migration of VSMCs from SHRs. Nevertheless, miR-155-5p deficiency afforded the opposite effects. Mechanistically, miR-155-5p directly targeted BACH1 to attenuate oxidative lesions in VSMCs under a hypertensive state. As such, our results shed new light on the role of miR-155-5p in VSMC migration, thus pinpointing miR-155-5p as a novel therapeutic candidate for hypertension-related vascular remodeling.

MiR-155-5p is documented as governing various functions of cells, including cardiovascular cells. The level of miR-155-5p is dysregulated in placental tissues from patients suffering from gestational hypertension [22]. The plasma miRNA-155-5p expression is significantly higher in patients with nocturnal hypertension [23]. As an inflammatory cytokine, TNF-α (tumor necrosis factor-α) is shown to induce the VSMC’s phenotypic switching from a contractile to a synthetic state, an effect that is blocked by treatment with an inhibitor of miR-155-5p [24]. MiR-155-5p is potentially therapeutic for hypertensive renal injury [25]. Increased miR-155-5p is partially involved in cold-exposure-aggravated pulmonary arterial hypertension [26]. It has been revealed that the miR-155-5p inhibitor remarkably inhibits the proliferation and migration of hypoxia-stimulated pulmonary artery smooth muscle cells [27]. In spite of these findings regarding miR-155-5p biology, the possible role and mechanism of miR-155-5p in VSMC migration and hypertensive vascular remodeling are unclear. In this study, our results showed that the sustained expression of miR-155-5p was able to attenuate the migration of VSMCs from SHRs, while the ablation of miR-155-5p potentiated the migratory abilities of SHR VSCMs, indicating that miR-155-5p acts as a suppressor in VSMC migration and vascular remodeling in hypertension.

Oxidative stress is a well-established driving force in the development of vascular remodeling in hypertension [28]. The scavenging of excessive reactive oxygen species (ROSs) can attenuate the migration of VSMCs [17]. BTB and CNC homology 1 (BACH1) is a ubiquitous master modulator that regulates the cellular oxidative stress response [29]. It has been suggested that BACH1 is a key negative regulator of Nrf2, a transcription factor that plays an important role in maintain intracellular oxidation homeostasis [30,31]. A substantial number of studies have disclosed that a deficiency in BACH1 is beneficial in a wide range of disorders [32], including hypertension [33]. We were intrigued to know whether miR-155-5p would restrain VSMCs’ oxidative stress and subsequent migration, via targeting BACH1. BACH1 being a target of miR-155-5p was predicted through the TargetScan website, and confirmed using a luciferase reporter assay. As expected, BACH1 was found to be a target gene of miR155-5p. Accordingly, our results showed that miR-155-5p directly inhibited the transcriptional and translational levels of BACH1, and suppressed the formation of ROSs in SHR VSMCs. In turn, the deletion of miR-155-5p augmented the expression of BACH1, and promoted the generation of superoxide anions, in conjunction with higher protein expressions of NOX2 and NOX4, two NAD(P)H oxidases serving as the main source of ROSs in cardiovascular cells. Synergistically, miR-155-5p overexpression, and BACH1 downregulation, further repressed the migration of VSMCs. Overall, the suppression of BACH1 by miR-155-5p was sufficient to reduce the migration and oxidative stress in SHR VSMCs.

## 5. Conclusions

MiR-155-5p regulated hypertension-induced VSMC migration by suppressing the expression of BACH1. The blockade of VSMC migration through the targeting of miR-155-5p expression in VSMCs may halt vascular remodeling in conditions (such as hypertension) with enhanced VSMC migration.

## 6. Limitations

In this study, we only explored the role of miR-155-5p in VSMC migration in vitro, and we only chose VSMCs from conduit vessels, not from resistant arteries; thus, it is crucial to further determine the role of miR-155-5p in hypertensive blood pressure remodeling in animal experiments. It should be considered that miR-155-5p may be detectable in other cell types, and we should further investigate the changes in human aortic VSMCs. Future studies should place emphasis on the cell-specific role of miR-155-5p in vascular remodeling, and it will likely be meaningful to demonstrate whether miR-155-5p is altered in resistant arteries or during circulation in both human and animal models of hypertension.

## Figures and Tables

**Figure 1 biomedicines-11-01679-f001:**
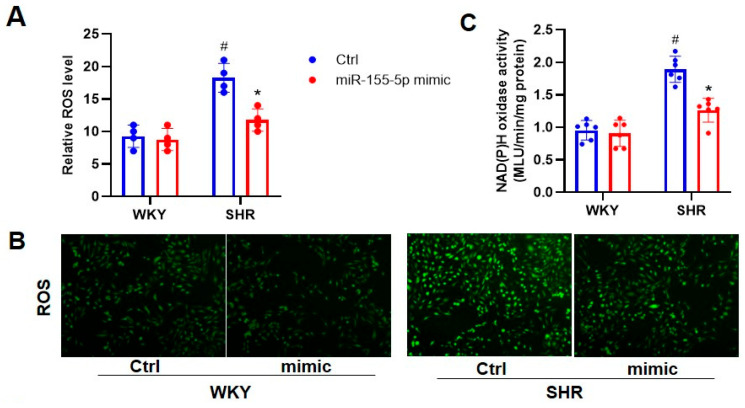
Effects of miR-155-5p mimics on oxidative stress in VSMCs from WKYs and SHRs. Measurement at 20 h after induction with 50 nmol/L normal control or 50 nmol/L miR-155-5p mimics. (**A**,**B**) ROS production. (**C**) NAD(P)H oxidase activities. (**D**) NOX2 and NOX4 protein levels. Values are expressed as mean ± SE. * *p* < 0.05 vs. Ctrl; # *p* < 0.05 vs. WKYs. *n* = 6 per group in (**C**), and *n* = 4 per group in (**A**,**B**,**D**).

**Figure 2 biomedicines-11-01679-f002:**
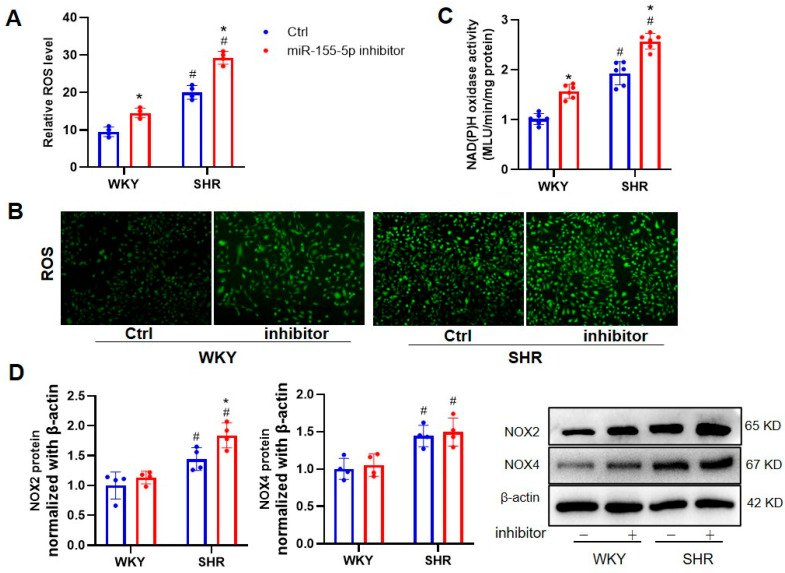
Effects of miR-155-5p inhibitors on oxidative stress in VSMCs from WKYs and SHRs. Measurement after a 24 h treatment with 100 nmol/L negative control or 100 nmol/L miR-155-5p inhibitor. (**A**,**B**) ROS levels. (**C**) NAD(P)H oxidase activity. (**D**) NOX2 and NOX4 protein expression. Values are expressed as mean ± SE. * *p* < 0.05 vs. Ctrl; # *p* < 0.05 vs. WKYs. *n* = 6 per group in (**C**), and *n* = 4 per group in (**A**,**B**,**D**).

**Figure 3 biomedicines-11-01679-f003:**
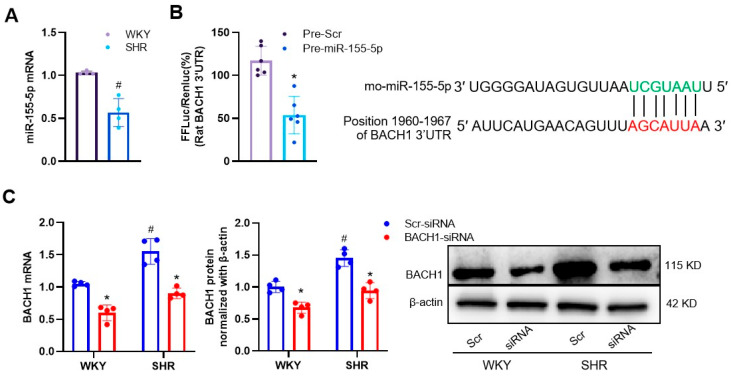
MiR-155-5p and BACH1 expressions in VSMCs of WKYs and SHRs. (**A**) Relative values of miR-155-5p levels measured with the qPCR method. (**B**) Prediction of the location of the miR-155-5p combination by TargetScanHuman, and dual-luciferase reporter assay showing that BACH1 is a target of miR-155-5p in VSMCs of WKYs. FFLuc is firefly luciferase; Renluc is Renilla luciferase. (**C**) Effects of BACH1 knockdown on BACH1 mRNA and protein expressions in VSMCs of WKYs and SHRs. The measurements were made after treatment with the control of lentivirus (Scr-siRNA, 80 MOI) or BACH1-siRNA (80 MOI) for 48 h. Values are mean ± SE. * *p* < 0.05 vs. Pre-Scr or Scr. # *p* < 0.05 vs. WKYs. *n* = 6 per group in (**B**), and *n* = 4 per group in (**A**,**C**).

**Figure 4 biomedicines-11-01679-f004:**
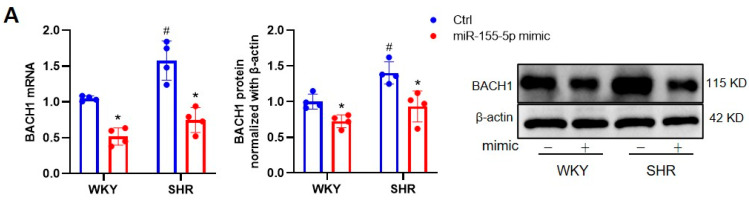
Effects of miR-155-5p on BACH1 expression in VSMCs. (**A**) Effects of miR-155-5p mimic on BACH1 mRNA and protein expressions in VSMCs. The measurements were carried out 24 h after negative control (Ctrl, 50 nmol/L) or miR-155-5p mimic (50 nmol/L) treatment. (**B**) Effects of miR-155-5p inhibitor on BACH1 mRNA and protein expressions in VSMCs. Measurements were performed 24 h after Ctrl (100 nmol/L) or miR-155-5p inhibitor (100 nmol/L) treatment. Values are mean ± SE. * *p* < 0.05 vs. Ctrl; # *p* < 0.05 vs. WKYs. *n* = 4 per group.

**Figure 5 biomedicines-11-01679-f005:**
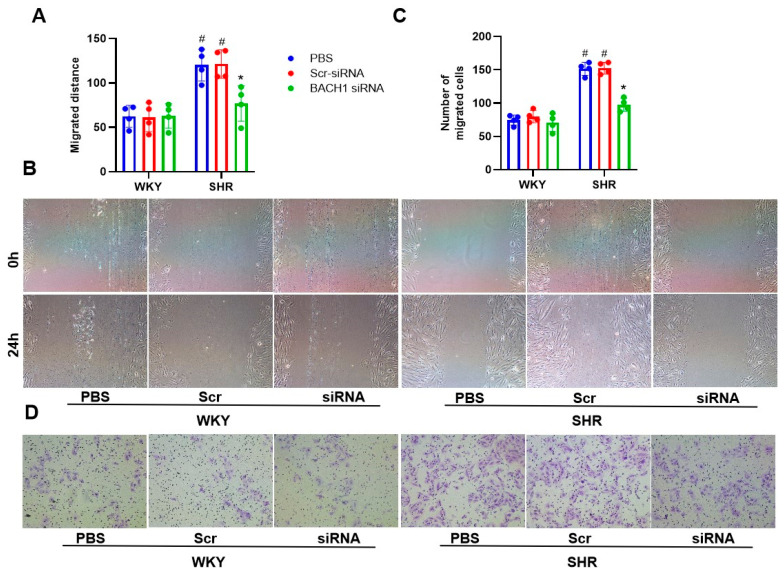
Effects of BACH1 knockdown on VSMC migration from WKYs and SHRs. Results of wound-healing assay (**A**,**B**) and Boyden chamber assay (**C**,**D**) at 48 h after challenge with PBS, 80 MOI Scr-siRNA, or 80 MOI BACH1-siRNA. Values are expressed as mean ± SE. * *p* < 0.05 vs. PBS; # *p* < 0.05 vs. WKYs. *n* = 4 per group.

**Figure 6 biomedicines-11-01679-f006:**
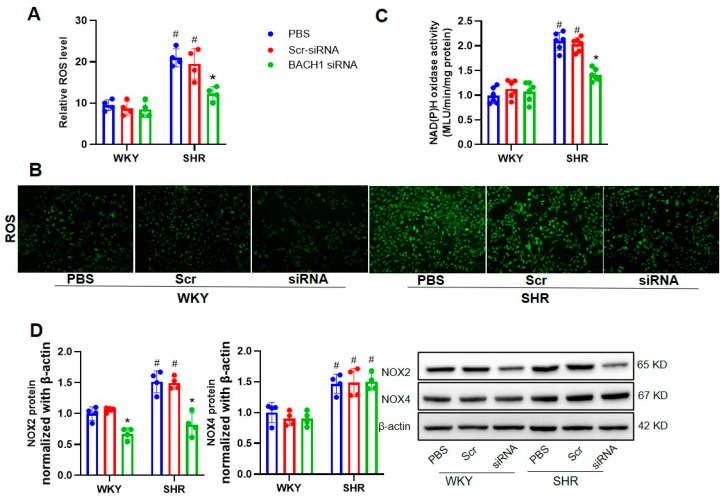
Effects of BACH1 knockdown on oxidative stress in VSMCs from WKYs and SHRs. Measurement at 48 h after induction with PBS, 80 MOI Scr-siRNA or 80 MOI BACH1-siRNA. (**A**,**B**) ROS levels. (**C**) NAD(P)H oxidase activity. (**D**) NOX2 and NOX4 protein expression. Values are expressed as mean ± SE. * *p* < 0.05 vs. PBS; # *p* < 0.05 vs. WKYs. *n* = 6 per group in (**C**), and *n* = 4 per group in (**A**,**B**,**D**).

**Figure 7 biomedicines-11-01679-f007:**
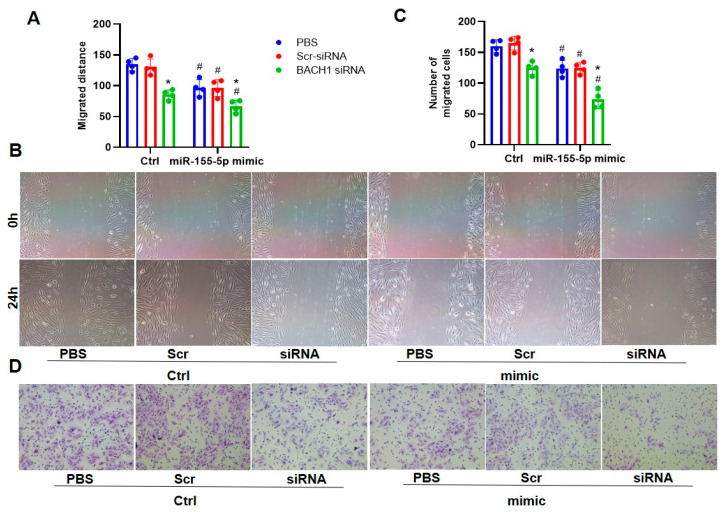
Effects of BACH1-siRNA on the miR-155-5p-mimic-induced VSMC migration in SHRs. Results of wound-healing assay (**A**,**B**) and Boyden chamber assay (**C**,**D**). SHR VSMCs were treated for 48 h with PBS, 80 MOI Scr-siRNA, or 80 MOI BACH1-siRNA, then for 24 h with 50 nmol/L normal control or 50 nmol/L miR-155-5p mimics. Values are expressed as mean ± SE. * *p* < 0.05 vs. PBS; # *p* < 0.05 vs. Ctrl. *n* = 4 per group.

**Figure 8 biomedicines-11-01679-f008:**
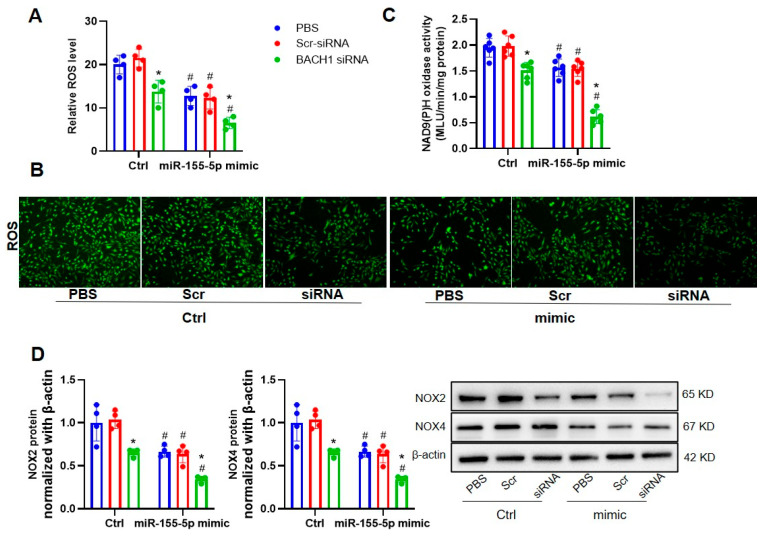
Effects of BACH1-siRNA on the miR-155-5p-mimic-induced oxidative stress in VSMCs from SHRs. SHR VSMCs were treated for 48 h with PBS, control lentivirus (Scr-siRNA, 80 MOI), or BACH1-siRNA (80 MOI), then for 24 h with 50 nmol/L normal control, or 50 nmol/L miR-155-5p mimics. (**A**,**B**) ROS production was detected. (**C**) NAD(P)H oxidase activity. (**D**) NOX2 and NOX4 protein expression. Values are mean ± SE. * *p* < 0.05 vs. PBS; # *p* < 0.05 vs. Ctrl. *n* = 6 per group in (**C**), and *n* = 4 per group in (**A**,**B**,**D**).

## Data Availability

Raw data for this study are available from the corresponding author upon reasonable request.

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
