# Peer review of "MiR-155-5p Attenuates Vascular Smooth Muscle Cell Oxidative Stress and Migration via Inhibiting BACH1 Expression"

_biomedicines, 2023, doi:10.3390/biomedicines11061679_

Round 1

Reviewer 1 Report

Please describe what “miR-155-5p inhibitors” are.

The labels for the images of Fig. 5 refer to “inhibitor” while the figure legend of Fig. 5 refers to “inhibitors”. Please clarify.

Line 144 states “By contrast, depletion of miR-155-5p …..” Does this “depletion” occur due to “inhibitor(s)”?

Wester blot images in Fig. 3D are overexposed and are of low quality.

Wester blot images in Fig. 4D are overexposed.

The BACH1 Western blot image in Fig. 5C is of low quality.

Fig. 5C is not described in the text and there is no figure legend for it.

N number?

No Panels B and D exist in Fig. 7 although they are mentioned in the legend.

Please define “Scr” of Scr-siRNA. 

For all the Western blotting quantification graphs, please use the values of the proteins of interest normalized with the house-keeping control proteins.

Author Response

  1. Please describe what “miR-155-5p inhibitors” a

RESPONSE: Thank you. The following sentence has been added into the method section to make it clear. “Scrambled sequences of the miR-155-5p mimic and inhibitor were used as negative controls to exclude the impact of the reagent or transfection protocol on the target protein.”

  1. The labels for the images of Fig. 5 refer to “inhibitor” while the figure legend of Fig. 5 refers to “inhibitors”. Please clarify.

RESPONSE: Good suggestion. ''Inhibitors'' has been revised as ''inhibitor'' in the manuscript.

  1. Line 144 states “By contrast, depletion of miR-155-5p …..” Does this “depletion” occur due to “inhibitor(s)”?

RESPONSE: Thank you. The following sentence has been replaced. “By contrast, miR-155-5p inhibitor not only induced the proliferation of WKY VSMCs, but also potentiated the migration distance and cell number of SHR VSMCs.”

  1. Wester blot images in Fig. 3D are overexposed and are of low quality.

RESPONSE: Good suggestion. We have revised them as your suggestion.

  1. Wester blot images in Fig. 4D are overexposed.

RESPONSE: Good suggestion. We have revised them as your suggestion.

  1. The BACH1 Western blot image in Fig. 5C is of low quality.

RESPONSE: Good suggestion. We have revised them as your suggestion.

  1. 5C is not described in the text and there is no figure legend for it.

RESPONSE: Thank you. We have revised them in the manuscript.

  1. N number?

RESPONSE: Thank you. The “n” is the number of per group, we have rewritten it in the figure legend section to make it clear.

  1. No Panels B and D exist in Fig. 7 although they are mentioned in the legend.

RESPONSE: Thank you.

  1. Please define “Scr” of Scr-siRNA. 

RESPONSE: Thank you. “Scr” of Scr-siRNA refers to “control lentivirus”, we have added it into figure legend to make it clear. 

  1. For all the Western blotting quantification graphs, please use the values of the proteins of interest normalized with the house-keeping control proteins.

RESPONSE: Thank you very much. We have revised them in the manuscript as your suggestion. 

  1. Please describe what “miR-155-5p inhibitors” a

RESPONSE: Thank you. The following sentence has been added into the method section to make it clear. “Scrambled sequences of the miR-155-5p mimic and inhibitor were used as negative controls to exclude the impact of the reagent or transfection protocol on the target protein.”

  1. The labels for the images of Fig. 5 refer to “inhibitor” while the figure legend of Fig. 5 refers to “inhibitors”. Please clarify.

RESPONSE: Good suggestion. ''Inhibitors'' has been revised as ''inhibitor'' in the manuscript.

  1. Line 144 states “By contrast, depletion of miR-155-5p …..” Does this “depletion” occur due to “inhibitor(s)”?

RESPONSE: Thank you. The following sentence has been replaced. “By contrast, miR-155-5p inhibitor not only induced the proliferation of WKY VSMCs, but also potentiated the migration distance and cell number of SHR VSMCs.”

  1. Wester blot images in Fig. 3D are overexposed and are of low quality.

RESPONSE: Good suggestion. We have revised them as your suggestion.

  1. Wester blot images in Fig. 4D are overexposed.

RESPONSE: Good suggestion. We have revised them as your suggestion.

  1. The BACH1 Western blot image in Fig. 5C is of low quality.

RESPONSE: Good suggestion. We have revised them as your suggestion.

  1. 5C is not described in the text and there is no figure legend for it.

RESPONSE: Thank you. We have revised them in the manuscript.

  1. N number?

RESPONSE: Thank you. The “n” is the number of per group, we have rewritten it in the figure legend section to make it clear.

  1. No Panels B and D exist in Fig. 7 although they are mentioned in the legend.

RESPONSE: Thank you.

  1. Please define “Scr” of Scr-siRNA. 

RESPONSE: Thank you. “Scr” of Scr-siRNA refers to “control lentivirus”, we have added it into figure legend to make it clear. 

  1. For all the Western blotting quantification graphs, please use the values of the proteins of interest normalized with the house-keeping control proteins.

RESPONSE: Thank you very much. We have revised them in the manuscript as your suggestion. 

Reviewer 2 Report

In this manuscript, the authors investigated the role of miR-155-5p in regulating oxidative stress and migration of primary VSMCs derived from thoracic aorta of WKY and SHR. The reviewer has the following concerns on the manuscript.

1.       Apparently, Figures 1-3 are just a repeat of published data in PMID: 32121598. The results are extremely similar. Moreover, PMID: 32121598 is not cited in this manuscript.

2.       The findings in this study will only be meaningful if the authors demonstrate that miR-155-5p are altered in resistant arteries and/or in the circulation in both human and animal models of hypertension. Although the authors claimed that ‘the level of miR-155-5p was lower in primary VSMCs from spontaneously hypertensive rats (SHR)’ (line 16), no data were presented in the manuscript.

3.       VSMCs were derived from thoracic aorta in this study. As hypertension was the focus of this study, the authors should justify why they chose VSMCs from conduit vessels but not from resistant arteries in this study. The authors did not mention whether all studies were carried out on primary VSMCs. If not, it should be noted that passaging of cells will cause alternations in gene expression and phenotype.

Minor English editing is recommended.

Author Response

  1. Apparently, Figures 1-3 are just a repeat of published data in PMID: 32121598. The results are extremely similar. Moreover, PMID: 32121598 is not cited in this manuscript.

RESPONSE: Good comments. The paper was cited in the manuscript.

  1. The findings in this study will only be meaningful if the authors demonstrate that miR-155-5p are altered in resistant arteries and/or in the circulation in both human and animal models of hypertension. Although the authors claimed that ‘the level of miR-155-5p was lower in primary VSMCs from spontaneously hypertensive rats (SHR)’ (line 16), no data were presented in the manuscript.

RESPONSE: Thank you very much. The following sentence has been revised into the limitations as your suggestion. “In this study, we only explored the role of miR-155-5p in VSMC migration in vitro and we only chose VSMCs from conduit vessels but not from resistant arteries in this study, thus it is crucial to further determine the role of miR-155-5p in hypertensive blood pressure remodeling in animal experiments and note. It should be meaningful to demonstrate whether miR-155-5p are altered in resistant arteries or in the circulation in both human and animal models of hypertension.” The data were presented in figure 5A.

  1. VSMCs were derived from thoracic aorta in this study. As hypertension was the focus of this study, the authors should justify why they chose VSMCs from conduit vessels but not from resistant arteries in this study. The authors did not mention whether all studies were carried out on primary VSMCs. If not, it should be noted that passaging of cells will cause alternations in gene expression and phenotype.

RESPONSE: Good suggestion. The following sentence has been revised into the limitations as your suggestion. “In this study, we only explored the role of miR-155-5p in VSMC migration in vitro and we only chose VSMCs from conduit vessels but not from resistant arteries in this study, thus it is crucial to further determine the role of miR-155-5p in hypertensive blood pressure remodeling in animal experiments. It should bear in mind that miR-155-5p may be detectable in other cell types.”

Reviewer 3 Report

In this interesting, paper, the authors explore the therapeutic effectiveness of investigating the exact role and action of miR-155-5p in VSMC migration by hypertension in vascular smooth muscle cells (VSMCs) isolated from from spontaneously hypertensive rats (SHR).

The authors observed protective effects after overexpression of miR-155-5p in VSMC while its depletion accelerated the migration and oxidative damage of VSMC from SHR. Dual luciferase reporter assay showed that miRNA-155-5p directly targeted BACH1 whereas miR-155-5p mimic inhibited BACH1 upregulation in SHR VSMCs. Finally the approved overexpression of miR-155-5p and knockdown of BACH1 had synergistic effects on the inhibition of VSMCs in hypertension.

The experimental approaches are sound and state of the art. In their results section and discussion, the authors conclude from their results that:  miR-155-5p attenuates VSMC migration and ameliorates vascular remodeling in SHR via 23 suppressing BACH1 expression.

However, there are considerable flaws in the experimental design and a number of major issues that should be addressed, as outlined in detail below.

Major Comment:

1.       The choice of applied cell culture model of rat VSMC hinders translation of the results  obtained from cell culture model to in vivo situation, since we now that the vascular origin of the VSMCs is important for the pharmacologically treatment of the calcification. VSMCs cultured from rat VSMCs react pharmacologically distinct as compared to cultured human aortic VSMCs. The adequate cell culture model should be human aortic VSMC. The authors should approve their major finding in human aortic VSMCs.

2.       The number of ethical approval should be listed

3.       Nearly all western blot bands are oversaturated the authors should provide bands with lover intensity (as shown in figure 8D) otherwise it is hard to see the differences between the different groups

Minor points:

1.       The authors should indicate from which passage to which passage their VSMCs were used for their experiments.

2.       In line 127 level of protein should be replaced by RNA and amount of RNA used for reverse transcription should be given

3.       Amount of miRNA used for reverse transcription and protein used for western blot should be also indicated

4.       Number of animals used for each experiment and per group should be written

In this interesting, paper, the authors explore the therapeutic effectiveness of investigating the exact role and action of miR-155-5p in VSMC migration by hypertension in vascular smooth muscle cells (VSMCs) isolated from from spontaneously hypertensive rats (SHR).

The authors observed protective effects after overexpression of miR-155-5p in VSMC while its depletion accelerated the migration and oxidative damage of VSMC from SHR. Dual luciferase reporter assay showed that miRNA-155-5p directly targeted BACH1 whereas miR-155-5p mimic inhibited BACH1 upregulation in SHR VSMCs. Finally the approved overexpression of miR-155-5p and knockdown of BACH1 had synergistic effects on the inhibition of VSMCs in hypertension.

The experimental approaches are sound and state of the art. In their results section and discussion, the authors conclude from their results that:  miR-155-5p attenuates VSMC migration and ameliorates vascular remodeling in SHR via 23 suppressing BACH1 expression.

However, there are considerable flaws in the experimental design and a number of major issues that should be addressed, as outlined in detail below.

Major Comment:

1.       The choice of applied cell culture model of rat VSMC hinders translation of the results  obtained from cell culture model to in vivo situation, since we now that the vascular origin of the VSMCs is important for the pharmacologically treatment of the calcification. VSMCs cultured from rat VSMCs react pharmacologically distinct as compared to cultured human aortic VSMCs. The adequate cell culture model should be human aortic VSMC. The authors should approve their major finding in human aortic VSMCs.

2.       The number of ethical approval should be listed

3.       Nearly all western blot bands are oversaturated the authors should provide bands with lover intensity (as shown in figure 8D) otherwise it is hard to see the differences between the different groups

Minor points:

1.       The authors should indicate from which passage to which passage their VSMCs were used for their experiments.

2.       In line 127 level of protein should be replaced by RNA and amount of RNA used for reverse transcription should be given

3.       Amount of miRNA used for reverse transcription and protein used for western blot should be also indicated

4.       Number of animals used for each experiment and per group should be written

Author Response

Major Comment:

  1. The choice of applied cell culture model of rat VSMC hinders translation of the results  obtained from cell culture model to in vivo situation, since we now that the vascular origin of the VSMCs is important for the pharmacologically treatment of the calcification. VSMCs cultured from rat VSMCs react pharmacologically distinct as compared to cultured human aortic VSMCs. The adequate cell culture model should be human aortic VSMC. The authors should approve their major finding in human aortic VSMCs.

RESPONSE: Thank you very much. In this study, we only explored the role of miR-155-5p in VSMC migration in vitro, it should bear in mind that miR-155-5p may be detectable in other cell types, and we should further approve the changes in human aortic VSMCs.

  1. The number of ethical approval should be listed

RESPONSE: Good comments. We have added it.

  1. Nearly all western blot bands are oversaturated the authors should provide bands with lover intensity (as shown in figure 8D) otherwise it is hard to see the differences between the different groups

RESPONSE: Thank you. We have revised them as your suggestion.

Minor points:

  1. The authors should indicate from which passage to which passage their VSMCs were used for their experiments.

RESPONSE: Thank you. We have added it into method section as your suggestion.

  1. In line 127 level of protein should be replaced by RNA and amount of RNA used for reverse transcription should be given

RESPONSE: Thank you very much. We have revised them as your suggestion.

  1. Amount of miRNA used for reverse transcription and protein used for western blot should be also indicated

RESPONSE: Thank you. The amount of miRNA used for reverse transcription and protein used for western blot has been added into the manuscript.

  1. Number of animals used for each experiment and per group should be written

RESPONSE: Good suggestion. The number of rats in each experiment and per group has been added into the manuscript.

Round 2

Reviewer 1 Report

  1. No Panels B and D exist in Fig. 7 although they are mentioned in the legend.
  1. For all the Western blotting quantification graphs, please use the values of the proteins of interest normalized with the house-keeping control proteins.

Author Response

Editor’s comments:

  1. No Panels B and D exist in Fig. 7 although they are mentioned in the legend.

RESPONSE: Thank you. We have added Panels B and D in Fig.7 in the revised manuscript.

  1. For all the Western blotting quantification graphs, please use the values of the proteins of interest normalized with the house-keeping control proteins.

RESPONSE: We actually used the values of the proteins of interest normalized with β-actin, thank you so much for your careful review, we have revised the figures in the updated manuscript.

Reviewer 2 Report

No further comments.

Author Response

Thank you very much.

Reviewer 3 Report

The authors have adequately addressed most, but not all, of the comments raised and in my opinion have significantly improved the manuscript. I think that the manuscript now merits publication in Biomedicines.

Author Response

Thank you very much.

Round 3

Reviewer 1 Report

.

Author Response

Dear editor: Comments and Suggestions for Authors: . .Therefore, I can not provide a point-by-point response to the reviewer’s comments. best regards pang